# Unexpected Cytological Detection of *Leishmania infantum* within the Secretion of a Canine Mammary Carcinoma

**DOI:** 10.3390/ani14192794

**Published:** 2024-09-27

**Authors:** Katrin Törner, Heike Aupperle-Lellbach, Elisabeth Müller, Torsten J. Naucke, Ingo Schäfer

**Affiliations:** 1LABOKLIN GmbH & Co. KG, Bad Kissingen, Steubenstrasse 4, 97688 Bad Kissingen, Germany; toerner@laboklin.com (K.T.); aupperle@laboklin.com (H.A.-L.); mueller@laboklin.com (E.M.); naucke@laboklin.com (T.J.N.); 2School of Medicine, Institute of Pathology, Technical University of Munich, Trogerstr. 18, 80333 Munich, Germany

**Keywords:** leishmaniasis, mammary gland, trans-mammary transmission, vector-borne infection, dog

## Abstract

**Simple Summary:**

A dog imported from Greece to Germany presented a mass in the mammary gland. Cytology of the secretion was performed, and besides the malignant epithelial cells, amastigotes of *Leishmania* (*L.*)*. infantum* were identified. Diagnoses of mammary carcinoma and canine leishmaniasis were confirmed by histopathology, PCR testing, and serology. The detection of *L. infantum* amastigotes in the mammary glands indicates a possible shedding by the milk and makes trans-mammary transmission to puppies theoretically possible as another route of *L. infantum* transmission apart from vectorial transmission by sand flies, blood transfusion, vertical transmission, and venereal transmission.

**Abstract:**

Mammary tumors are one of the most common neoplasms in female dogs, and cytology represents a non-invasive diagnostic method. The protozoal pathogen *Leishmania* spp. was previously demonstrated in canine mammary glands. An eight-year-old, female-spayed Doberman was imported from Crete, Greece, three years before the first presentation. The dog was presented due to a mammary tumor two years after adoption. The clinical examination revealed fever and weight loss. Smears of the mammary secretion were investigated cytologically. Multiple atypical epithelial cells with moderate to marked criteria of malignancy were detected. Furthermore, amastigotes were visible intra- and extracellularly. The diagnosis of *L. infantum* infection was based on a positive PCR out of the cytological smear, and a positive serology. Mammary carcinoma and granulomatous inflammation with amastigotes were confirmed by histopathology. We aimed to provide a complete report of cytological, histopathological, hematological, and biochemistry findings in a dog with *L. infantum* in the mammary glands with focus on trans-mammary pathogen transmission as a potential alternative way of spreading *Leishmania* infections. Canine leishmaniasis should be a potential differential diagnosis in dogs with lesions and/or inflammation in the mammary glands and a history of presence in areas endemic for *L. infantum*, especially the Mediterranean in Europe.

## 1. Introduction

Mammary tumors are considered the most common neoplasm in female dogs [1]. Masses in the canine mammary gland may be of inflammatory and/or neoplastic origin. In general, histopathology is considered the gold standard for differentiation between inflammatory and neoplastic lesions and differentiation between benign and malignant masses in cases of neoplasia. Cytology as a cost-effective and non-invasive diagnostic method provides satisfactory sensitivity (88%) and specificity (96%) for diagnosis of malignancy in canine mammary tumors compared to histopathology [2], with comparable results in human medicine (87–97% sensitivity, 99–100% specificity) [3,4,5,6,7]. However, cell morphology may vary in different areas of the tumor, leading to false negative or false positive results, especially in highly proliferative lesions and if the cytological specimen is not adequately representative [8,9]. Low diagnostic accuracy of cytology compared to histopathology is also demonstrated in case of necrosis and inflammation in mammary tumors due to relatively marked cellular atypia misinterpreted as carcinoma in cytology [2,8]. In dogs, inflammation of mammary tissue and the regional skin may be caused by bacteria [10,11], fungi [12,13], prototheca [14], or nematodes [15,16,17], or may be secondary to neoplastic growth. 

Additionally, the protozoal pathogen *Leishmania* spp. was demonstrated in mammary glands of dogs [18] and cats [19], as well as in humans diagnosed with breast and skin cancers [20,21,22]. In general, leishmaniasis is considered a global emerging disease, with phlebotomine sandflies as primary vectors and dogs as the main pathogen reservoir [23,24]. In addition to vector transmission, blood transfusion [25,26,27,28,29], vertical transmission [30,31], and venereal transmission [31,32] have been proven to be other potential sources for canine infections. In human medicine, skin cancers predominantly develop at sites of previous leishmanial lesions or scars, and leishmaniasis is observed as an opportunistic infection in humans, especially those with hematological cancers [22].

In dogs, *Leishmania* (*L.*) *infantum* was demonstrated in the mammary glands in 13/20 individuals (65%) with visceral leishmaniasis, of which nine (45%) were diagnosed by histopathology [18]. Amastigotes were observed in the mammary tissue in female dogs with intra- and interlobular distribution as well as around and/or in the ductal lumen associated with an intramammary mononuclear granulomatous inflammation [18]. Histopathology was dominated by macrophages, plasma cells, and lymphocytes, whereas neutrophilic and eosinophilic granulocytes were considered rare [18]. In natural and experimental infections with *L. infantum* and *L. chagasi*, amastigotes can be detected in the cytoplasm of macrophages, which are distributed systematically, and can reach the female reproductive system in visceral leishmaniasis [18,33,34]. 

Serum progesterone levels in the mammary gland of infected female dogs can influence the maintenance of an immunosuppressive cell profile and, therefore, promote the persistence of *L. infantum* amastigotes in the mammary glands [35]. However, no correlation between the parasite load and levels of progesterone has been described [35]. The hormonal influence in the mammary gland can modulate leukocytes to be less protective [35]. 

The detection of amastigotes in the lumen of the mammary glands indicates a possible shedding of *L. infantum* by the milk, and predominantly granulomatous mastitis is a frequent clinical finding [18]. One hypothesis is that puppies could be infected by ingesting amastigotes together with milk, and amastigotes may be able to enter the bloodstream through lesions in the mucosa of the upper digestive tract, as survival of amastigotes is unlikely in acid–pepsin digestion in the stomach [18]. 

The aim of this case report is to provide a complete report of cytological, histopathological, hematological, and biochemistry findings in a dog with *L. infantum* in the mammary glands to place focus on trans-mammary pathogen transmission as a potential alternative means of spreading *Leishmania* infections. Additionally, the value of cytology should be highlighted in the presented case.

## 2. Materials and Methods

An eight-year-old, female-spayed Dobermann dog with a body weight of 26 kg was imported from Crete in Greece to Germany three years before the first presentation. Two years after adoption, the dog was presented because of a mass in the mammary region. The clinical examination revealed fever with a rectal temperature of 39.4 °C and a weight loss of 4 kg bodyweight (15%) compared to a previous visit three months ahead. A hemodiluted and milky secretion was recognized out of the lumen of the mammary glands. 

Two air-dried smears prepared from secretion of the mammary gland of the dog were stained with a Diff-Quick rapid stain (LABOR + TECHNIK Eberhard Lehmann GmbH, Berlin, Germany) according to the standard operation procedure in the laboratory LABOKLIN with the aim of differentiating an inflammatory lesion (mastitis) from a neoplasm.

A tissue sample from two mammary complexes measuring 22.4 × 6.4 × 4.1 cm was submitted for histopathology. Formalin-fixed material was cut into pieces according to the guidelines of Kamstock et al. [36]. Two 0.4 cm and one 2.4 cm masses were identified. Representative localizations of the masses and the lymph node (1.4 cm in diameter) were embedded in paraffin wax and cut at 4–5 µm thickness to then be stained routinely with hematoxylin–eosin (HE).

A complete blood count (CBC, Sysmex XN-V analyzer, Sysmex Deutschland, Norderstedt, Germany), a biochemical profile (Cobas 8000, Roche Deutschland Holding GmbH, Mannheim, Germany), and a serum protein capillary electrophoresis (Sebia Minicap, Sebia, Mainz, Germany) test were performed in the LABOKLIN laboratory (Bad Kissingen, Germany). TaqMan^®^ real-time PCR testing for the detection of *L. infantum* was carried out at LABOKLIN using a LightCycler96 (Roche Diagnostics, Mannheim, Germany). Cycle threshold (cq) values below 35 were considered positive. Each PCR run included a negative and a positive control as well as an extraction control in each sample, to check for nucleic acid extraction and PCR inhibition (DNA/RNA Process Control Detection Kit, Roche Diagnostics GmbH, Mannheim, Germany). Testing for *Leishmania* spp. by PCR was performed using one of the submitted cytological slides, in which presence of amastigotes was noted microscopically (target: kinetoplast minicircle DNA; primer: 5′—AAC TTT TCT GGT CCT CCG GGT AG—3′, 5′—ACC CCC AGT TTC CCG CC—3′; probe: 5′—FAM—AAA AAT GGG TGC AGA AAT—NFQMGB—3′ [37]). For serological detection of *Leishmania* spp., enzyme-linked immunosorbent assay (ELISA) testing (NovaTec VetLine Leishmania ELISA, Immundiagnostica GmbH, Dietzenbach, Germany, >11 LE positive) was used on serum. 

The dog was tested for further coinfections using a “Canine Southern European travel profile” in the laboratory LABOKLIN, including PCR testing for *Hepatozoon canis* (TaqMan^®^ real-time PCR, in-house test, amplifying a ~664-base pair (bp) fragment of the 18S ribosomal [rRNA] gene), *Anaplasma platys* (TaqMan^®^ real-time PCR, in-house test), and *Dirofilaria* spp. (TaqMan^®^ real-time PCR, in-house test for detection of microfilariae); we also carried out serological antibody detection for *Babesia canis* (ELISA, Babesia ELISA Dog, Afosa, Blankenfelde-Mahlow, Germany), *Ehrlichia canis* (ELISA, Ehrlichia ELISA Dog, Afosa, Blankenfelde-Mahlow, Germany), and *Rickettsia* spp. (IFAT, RICKETTSIA CONORII IFA SLIDE, Viracell, Granada, Spain), as well as *Dirofilaria immitis* antigen-testing (FASTest^®^ HW Antigen, MegaCor GmbH, Hörbranz, Austria).

## 3. Results

Testing for vector-borne pathogens with an unknown panel was performed in Greece before adoption, revealing negative results for each pathogen. After adoption, the dog was presented several times with swollen joints in different locations, lameness, dermatological signs, panniculitis, pruritus, and gastrointestinal disorders. A clinical diagnosis of polyarthritis and allergy was raised without performing further diagnostic workup due to financial restrictions by the owner. The dog was on a low dosage of prednisolone (Prednicortone 5 mg tablets for use in dogs and cats, Dechra Regulatory B. V., 0.2 mg/kg orally twice daily) and, if necessary, novaminsulfone (Metamizol Hexal 500 mg tablets, Hexal AG, Holzkirchen, Germany, 40 mg/kg up to three times daily).

Cytologically, the smears were of moderate cellularity with high numbers of erythrocytes and macrophages as well as low to occasional numbers of small mature lymphocytes and non-degenerated neutrophils (Figure 1). Additionally, multiple epithelioid cells were seen in variably sized cohesive clusters or spread individually. Anisocytosis and anisokaryosis were moderate. The chromatin pattern was coarse, with sometimes prominent nucleoli. Mitotic figures were occasionally detected. The criteria of malignancy lead to the most likely diagnosis of a mammary adenocarcinoma. 

Unexpectedly, multiple 1.5 to 2.0 × 2.5 to 5.0 µm sized structures with a round nucleus and characteristic bar-shaped kinetoplasts, consistent with amastigotes of *Leishmania* spp., were recognized intracellularly in macrophages and extracellularly (Figure 1). 

The cytological finding of amastigotes in cytology was verified by positive PCR testing for *L. infantum* out of one of the submitted smears. PCR testing out of the EDTA blood revealed a positive result with 343 *Leishmania*/mL, representing a high parasite load (Table 1). Serological testing for *L. infantum* was positive (19 LE, <11 LE considered negative). 

The histopathological investigation showed poor tissue preservation with moderate autolysis. Three low-malignant complex carcinomas were diagnosed. They originated from the glandular and ductal epithelium and the myoepithelial of the mammary. The growth pattern was predominantly tubular to tubulopapilliform. The tumor cells were well to moderately differentiated and showed mild to moderate pleomorphism. There were single mitoses and nuclear atypia. The neoplasms grew infiltrative, but no vascular invasions were detectable. Multifocally, there was congestion of secretion with cystic dilation of the ducts. The margins were clean. Additionally, there was a diffuse mild to moderate mixed-cell interstitial inflammation with the involvement of numerous macrophages. In the macrophages, numerous intracellular pathogen structures are recognizable, whose morphology is consistent with *Leishmania*. Focally, intraluminal macrophages with *Leishmania* were found (Figure 2A). No metastases of the complex carcinomas were detectable in the lymph node, but lots of macrophages with intracellular *Leishmania* and mild hemosiderosis were visible (Figure 2B).

The CBC revealed mild thrombocytopenia as well as mild absolute neutrophilia and monocytosis with eosinopenia and lymphopenia (Table 1). In the biochemistry panel assessment of serum, moderate hyperproteinemia with hyperglobulinemia and moderate elevation of liver parameters were seen (Table 1). The mild hemolysis in the serum most likely caused the hyperkalemia, hyperphosphatemia, and elevation of urea (Table 1). Polyclonal peaks in the alpha-2, beta-2, and gamma sections were seen in the serum protein capillary electrophoresis (Figure 3). 

Polyclonal peaks in the alpha 2, beta 2, and gamma sections were seen in the serum protein capillary electrophoresis (Figure 3). 

The screening for potential coinfections revealed negative results except for positive results in serological testing for *Rickettsia* spp. (1:512, <1:128 considered negative) and *Babesia canis* (55.19 TE, <19 TE considered negative).

In summary, the investigation of the secretion of a mammary mass resulted in the unexpected diagnosis of a severe leishmaniasis alongside the complex mammary carcinomas.

## 4. Discussion

In the present case, cytological investigation of the secretion of a mammary nodule revealed an adenocarcinoma as a malignant tumor, and unexpectedly, leishmaniasis was diagnosed as well. Leishmaniasis was confirmed by histology and PCR. This highlights the necessity of considering leishmaniasis a potential differential diagnosis even in cases of previously negative test results and the need to ask about any stays abroad in countries endemic for *Leishmania* spp. This is linked to the disease’s prolonged incubation period, with the development of clinical diseases occurring up to 7–10 years after initial contact with the pathogen [38]. Additionally, the value of cytology should be highlighted due to the unexpected finding of amastigotes in the mammary tumor in the reported case. Although cortisone was not applied in an immunosuppressive dosage, its application may have contributed to the acute-on-chronic course of leishmaniasis alongside the adenocarcinoma, as immunosuppression is well known as a risk factor for clinical onset of leishmaniasis in human medicine [20,21,22].

Associations between leishmaniasis and neoplasia are evaluated in human medicine with different pathogenesis: leishmaniasis can mimic a malignant disorder in squamous cell carcinoma, T-cell and B-cell lymphoma, oral and intranasal neoplasms, and granulomas [39]; leishmaniasis can follow chemotherapy in various malignancies; leishmaniasis can coexist with neoplasia in immunocompromised patients (predominantly due to HIV); and *Leishmania* spp. can be directly involved in the development of cancer, as especially demonstrated in the skin and mucous membranes [22]. It is hypothesized that a Th2 cytokine microenvironment, as shown in progressive leishmaniasis, may promote tumor cell proliferation and vice versa [39]. The promotion of cancer development by *Leishmania* spp. has been demonstrated in immunocompromised (esp. HIV) and immunocompetent people [39]. In cats, one case report has dealt with feline leishmaniasis with a concurrent nasal squamous cell carcinoma [40]. In dogs, visceral leishmaniosis is associated with transmissible venereal tumors [41,42,43] and, in individual cases, with lymphoma [44,45], fibrosarcoma [45], or adrenocortical adenoma [45]. In the presented dog, leishmaniasis was an incidental finding by evaluating the cytological smears taken to differentiate between an inflammatory lesion and a neoplastic disorder in the mammary glands. An association of the carcinoma diagnosed via histopathology with leishmaniasis can only be hypothesized.

In general, limited data are available concerning the findings of amastigotes of *Leishmania* in mammary glands in dogs and cats. Thus far, only individual case reports are available, describing detection of *L. infantum* amastigotes in natural infections in the lumen of mammary glands in dogs without providing any further details regarding clinical signs or hematological and biochemistry abnormalities [18]. In cats, only one case report has been published describing a *L. donovani* infection in the mammary gland [19]. This study’s report adds knowledge regarding clinical signs, cytological abnormalities, and hematological as well as biochemistry results in dogs with *L. infantum* in the mammary glands. Therefore, leishmaniasis could be a potential differential diagnosis in dogs and cats with lesions and/or inflammation in the mammary glands and a history of presence in areas endemic for *L. infantum*, especially the Mediterranean in Europe. 

Due to the zoonotic potential of *L. infantum*, it is important to gain knowledge regarding the pathogenesis of canine leishmaniasis, as dogs are considered one of the major pathogen reservoirs [46]. The possibility of trans-mammary transmission cannot be ruled out and may contribute to the spread of leishmaniasis. Therefore, and because of the venereal as well as vertical transmission of *L. infantum* as additional routes of the spread of this pathogen [30,32], castration should be recommended in any female or male dog imported from endemic areas to other countries in which *L. infantum* is not yet endemic.

The cytological findings—with predominantly granulomatous inflammation and detection of numerous intra- and extracellular amastigotes—are in accordance with histopathological findings in another study [18]. 

Canine leishmaniasis was diagnosed by positive PCR testing of the EDTA blood and cytological smears with numerous intra- and extracellular amastigotes; hematological (thrombocytopenia) and biochemistry (hyperproteinemia with hyperglobulinemia) results consistent with leishmaniasis [47,48], and positive serology. The anamnesis was also consistent with leishmaniasis, this study’s dog having been imported from Greece, which is known to be a highly endemic area for *L. infantum* [47]. Polyarthritis may have been present in parallel with the *L. infantum* infection, although it is also known to be a potential clinical sign in dogs with leishmaniasis [47]. The application of glucocorticoids and/or a stress response may have caused the lymphopenia and eosinopenia, whereas the neutrophilia and monocytosis are consistent with leishmaniasis.

The positive serology for *Rickettsia* spp. and *Babesia canis* most likely did not contribute to the course of disease. Both positive serological results are indicative of contact with the pathogen in the past but do not prove an acute infection. In Germany, 78% of dogs were found to be serologically positive for *Rickettsia* spp. without any correlation between clinical status, location, and seropositivity, suggesting that rickettsial infection is related neither to the health condition of the dogs nor to their living in Germany [49].

## 5. Conclusions

In conclusion, canine leishmaniasis should be a potential differential diagnosis in dogs with lesions and/or inflammation in the mammary glands and a history of presence in areas endemic for *L. infantum*, especially the Mediterranean in Europe. The cytological and/or histopathological detection of amastigotes of *L. infantum* is possible in the lumen of canine mammary glands. Cytological and/or histopathological results should be confirmed by PCR testing. 

The detection of *L. infantum* amastigotes in the mammary glands indicates a possible shedding of *L. infantum* by the milk. Alongside vectorial transmission, blood transfusion, vertical transmission, and venereal transmission, the possibility of trans-mammary transmission cannot be ruled out and may contribute to the spread of leishmaniasis. However, this must be further investigated.

## Figures and Tables

**Figure 1 animals-14-02794-f001:**
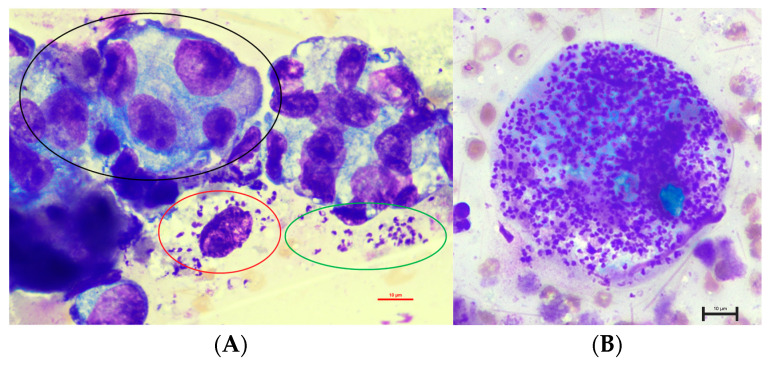
Cytology of the fluid taken from the lumen of the mammary gland in an eight-year-old, female-spayed Dobermann infected with *Leishmania infantum* with epithelial cells with marked atypia ((**A**), black circle) as well as intracellular ((**A**): red circles in a macrophage, (**B**): numerous amastigotes in a macrophage) and extracellular amastigotes ((**A**): green circle) (1000× magnification).

**Figure 2 animals-14-02794-f002:**
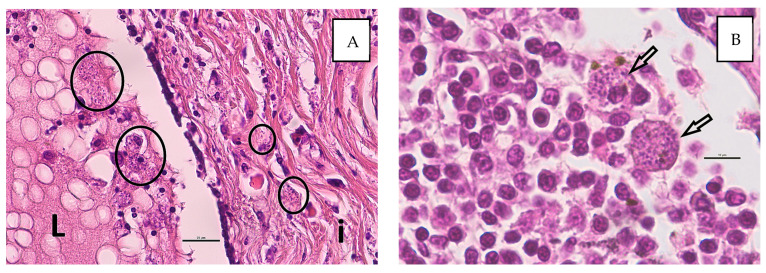
Histological detection of *Leishmania* spp. in macrophages ((**A**), circles) within the interstitium (i) and intraluminal (L) in the secretion of the mammary tissue (**A**) and within the regional lymph node ((**B**), arrows). (HE stain, bars: (**A**) 25 µm, (**B**) 10 µm).

**Figure 3 animals-14-02794-f003:**
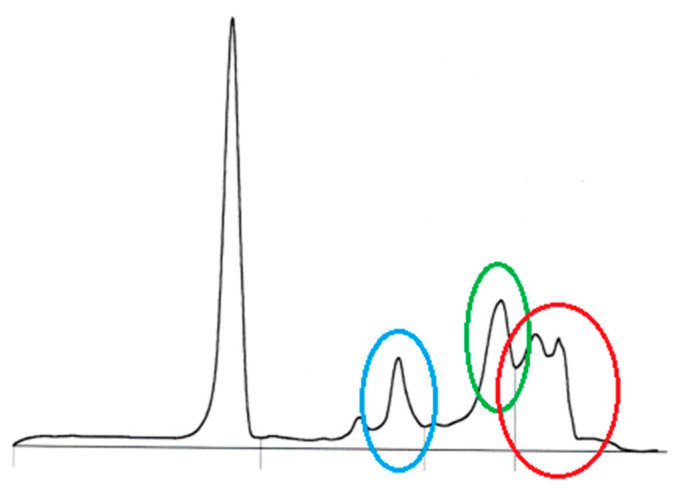
Serum protein capillary electrophoresis in an eight-year-old, female-spayed Dobermann infected with *Leishmania infantum* with polyclonal peaks in the alpha 2 (blue circle) and gamma sections (red circle), and a polyclonal peak in the beta 2 section with beta–gamma bridging (green circle); Sebia Minicap (Sebia, Mainz, Germany).

**Table 1 animals-14-02794-t001:** Complete blood count, biochemical parameters, and parasitological status in an eight-year-old, female-spayed Dobermann infected with *Leishmania infantum* performed by the laboratory LABOKLIN (Bad Kissingen, Germany).

Parameter	Reference Interval	Values
**Complete Blood Count ^1^**		
Red blood cells (×10^12^/L)	5.5–8.5	6.52
Hemoglobin (g/L)	150–190	148
Hematocrit (L/L)	0.44–0.52	0.46
White blood cells (×10^12^/L)	6.0–12.0	10.9
Segmented (×10^9^/L)	3.0–9.0	9.2
Bands (×10^9^/L)	<0.6	0.0
Lymphocytes (×10^9^/L)	1.0–3.6	0.9
Eosinophils (×10^9^/L)	0.04–0.6	0.0
Monocytes (×10^9^/L)	0.04–0.5	0.8
Hypochromasia	Negative	Negative
Anisocytosis	Negative	Negative
Platelets (×10^9^/L)	150–500	135
**Biochemistry ^2^**		
Alpha-Amylase (U/L)	<1650.0	335
DGGR-lipase (U/L)	<120.0	101.9
Fructosamine (µmol/L)	<374	270.3
Triglycerides (mmol/L)	<3.9	3.18
Cholesterol (mmol/L)	3.1–10.1	7.5
Bilirubin (µmol/L)	<3.4	2.4
Alkaline phosphatase (U/L)	<147	620
Glutamate dehydrogenase (U/L)	<8.0	66.9
Gamma-glutamyl transferase (U/L)	<10.0	43.0
Alanine transaminase (U/L)	<88.0	678.9
Aspartate aminotransferase (U/L)	<51.0	67.2
Creatin kinase (U/L)	<200.0	139.0
Total Protein (g/L)	54.0–75.0	90.1
Albumin (g/L)	25.0–44.0	34.9
Globulin (g/L)	<45.0	55.2
Urea (mmol/L)	3.3–8.3	8.5
Creatinine (µmol/L)	<125	82.0
Phosphorus (mmol/L)	0.7–1.6	1.8
Magnesium (mmol/L)	0.6–1.3	0.8
Calcium (mmol/L)	2.3–3.0	2.6
Sodium (mmol/L)	140–155	146
Potassium (mmol/L)	3.5–5.1	5.4
Iron (µmol/L)	15–45	36.4
***Leishmania* testing**		
*Leishmania* ELISA ^3^	<11 LE	19
*Leishmania* qPCR (/mL)	-	343 (positive)

Segmented = segmented neutrophilic granulocytes; Bands = banded neutrophilic granulocytes; ELISA = enzyme-linked immunosorbent assay; qPCR = quantitative polymerase chain reaction; ^1^ Sysmex XN-V analyzer, Sysmex Deutschland, Norderstedt, Germany; ^2^ Cobas 8000, Roche Deutschland Holding GmbH, Mannheim, Germany; ^3^ NovaTec VetLine Leishmania ELISA, Immundiagnostica GmbH, Dietzenbach, Germany.

## Data Availability

The original contributions presented in the study are included in the article, further inquiries can be directed to the corresponding author.

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
