# Peer review of "Unexpected Cytological Detection of Leishmania infantum within the Secretion of a Canine Mammary Carcinoma"

_animals, 2024, doi:10.3390/ani14192794_

Round 1

Reviewer 1 Report

Comments and Suggestions for Authors

This is a very well written report of a finding of Leishmania infantum amastigotes in a 8-year old Dobermann imported from Greece  with mammary adenocarcinoma. This finding is of interest in terms of other routes of leishmania transmission as well as the eventual association between leishmania and neoplasic disorders.

Material and methods:  In order to increase manuscript comprehension,  I would strongly recommend to include dog anamnesis and clinical history ( sentence 127-128 as well as the  130-142) into the Methodology section instead of the Results section. Otherwise, this manuscript could be presented as a case report instead of a research study.

Results: It could be very helpful for understanding the importance of this finding to get some information about the follow-up of this dog once it was diagnosed:   Was this dog treated ? Surgical and Medical treatment? Any follow-up? Recovered? Relapsed?

Discussion: I missed some discussion about the potential association between Leishmania infantum and malignant disorders. Some other reports have already been published about the eventual association between leishmania and carcinoma:

Schwing A, Pomares C, Majoor A, Boyer L, Marty P, Michel G. Leishmania infection: Misdiagnosis as cancer and tumor-promoting potential. Acta Trop. 2019 Sep;197:104855. doi: 10.1016/j.actatropica.2018.12.010. Epub 2018 Dec 7. PMID: 30529443.

Maia C, Sousa C, Ramos C, Cristóvão JM, Faísca P, Campino L. First case of feline leishmaniosis caused by Leishmania infantum genotype E in a cat with a concurrent nasal squamous cell carcinoma. JFMS Open Rep. 2015 Jul 1;1(2):2055116915593969. doi: 10.1177/2055116915593969. PMID: 28491373; PMCID: PMC5362010.

Reviewer 2 Report

Comments and Suggestions for Authors

Leishmaniasis is a global emerging disease which can be demonstrated in animals and humans. This study investigated the secretion of a mammary mass from a female dog though different methods, and unexpected diagnosis of a severe leishmaniasis beside the complex mammary carcinomas. However, something special should be focused on and written in more words. The text should be improved as well.

Specific comments:

1.     Abstract. What’s the purpose of this study? Please supply it which will be easier for readers to know your research quickly.

2.     Keywords. It is suggested to show “leishmaniasis” firstly, and use “mammary glands” as keyword as well.

3.     Line 53-71. Leishmaniasis has been reported in mammary glands of dogs as there are also some papers that have demonstrated it. So what’s the meaning of Leishmania infantum detected in this case? Is there any difference? Which part is special could be focused on? Something novelly are more likely to attract readers' interest and should be shown in more words in Discussion. 

4.     Line 127-128. The source of dog should be moved to the “2. Materials and Methods”.

5.     There are no solid data to support the conclusion. What is the relationship between canine leishmaniasis and mammary glands? And is there the connection?

6.     There are some grammar and spelling mistakes which should be check and correct.

(1)   Latin names should be italic. For example, “L. infantum” in line 31. Please check the whole text and correct.

(2)   Although “FAM” and “NFQMGB” are common, it is necessary to show the full names as well.

(3)   Table should be present as three-line.

Comments on the Quality of English Language
